# A Review of Immunotherapy in Renal Cell Carcinoma: Current Landscape and Future Directions

**DOI:** 10.3390/cancers17193139

**Published:** 2025-09-26

**Authors:** Supriya Peshin, Adit Dharia, Nagaishwarya Moka, William Paul Skelton

**Affiliations:** 1Department of Internal Medicine, Norton Community Hospital, Norton, VA 24273, USA; 2Department of Internal Medicine, HCA Florida Oak Hill Hospital, Brooksville, FL 34613, USA; 3Department of Hematology and Oncology, Lincoln Memorial University, Harrogate, TN 38105, USA; 4Department of Medicine, Division of Hematology and Oncology, University of Virginia Comprehensive Cancer Center, Charlottesville, VA 22908, USA

**Keywords:** renal cell carcinoma, immunotherapy, immune checkpoint inhibitors, tyrosine kinase inhibitors, PD-1, CLTA-4, belzutifan, advanced RCC

## Abstract

This article focuses on the evolving role of immunotherapy in renal cell carcinoma (RCC), beginning with preclinical insights and progressing through landmark clinical trials that led to current treatment approvals. Immune checkpoint inhibitors (ICIs), particularly in combination with tyrosine kinase inhibitors (TKIs) or other ICIs, have significantly improved outcomes for patients with advanced RCC. We summarize pivotal trials such as CheckMate 214, KEYNOTE-426, and CheckMate 9ER, which established the role of IO-based combinations. The article also discusses mechanisms of resistance, primary, adaptive, and acquired, and outlines tumor-intrinsic, host-related, and microenvironmental contributors. To overcome these challenges, we review emerging strategies including novel checkpoint targets (LAG-3, TIGIT), cytokine-based therapies, personalized mRNA vaccines, CAR T-cell approaches, and modulation of the gut microbiome. Additionally, we highlight ongoing trials such as LITESPARK (belzutifan), PEDIGREE (adaptive therapy), and PROBE (cytoreductive nephrectomy) that may shape future practice. This review provides a comprehensive overview of immunotherapy’s current landscape and future direction in RCC.

## 1. Introduction

Renal cell carcinoma (RCC) arises from the renal tubular epithelium and accounts for over 80% of all kidney cancers [1]. In 2024, it was estimated that 81,610 new cases of RCC would be diagnosed, representing 4.1% of all new cancer cases. Additionally, the projected number of deaths due to RCC in the same year was 14,390, which constitutes 2.4% of all cancer-related deaths [2]. The primary risk factors for RCC include tobacco use, obesity, hypertension, Von Hippel–Lindau syndrome, acquired cystic kidney disease, and genetic predisposition [3]. The higher incidence of RCC observed in Europe and North America compared to Asia and Africa is largely attributed to the greater accessibility and utilization of advanced diagnostic imaging in Western countries [4].

Clear cell RCC (ccRCC) accounts for approximately 80% of cases, followed by papillary RCC (15%), both originating from the proximal tubule. Less common subtypes include chromophobe, collecting duct, medullary, and translocation-associated RCC [5]. These vary in origin, clinical behavior, and prognosis, with medullary and sarcomatoid variants being particularly aggressive and affecting younger patients [6,7].

Various treatment modalities have been developed to inhibit tumor progression and metastasis in RCC. Current treatment strategies for RCC include surgical interventions such as nephrectomy, systemic therapies including ICI and targeted agents, and localized approaches such as thermal or cryoablation [8]. Radiation therapy is typically reserved for palliation in select metastatic or symptomatic cases [8]. Although many RCC cases are resectable at diagnosis, a significant proportion of patients present with advanced or metastatic disease, where nephrectomy may not be beneficial or feasible [9]. Furthermore, it is important to recognize that metastatic progression can occur in over one-third of patients, even after the surgical removal of the primary tumor [6].

Historically, treatment options for advanced RCC included tyrosine kinase inhibitors (TKIs), chemotherapy, and radiation, which have shown limited durability and low complete response rates [1,10]. In contrast, immunotherapy has become a cornerstone of modern cancer treatment, with increasing evidence supporting its role in improving long-term survival and disease control in RCC. Among the most studied and widely used immunotherapies are ICIs, which target specific transmembrane proteins that cancer cells exploit to evade immune surveillance. By inhibiting these proteins, ICIs restore the immune system’s ability to recognize and attack tumors, thereby preventing disease progression and metastasis [8].

Despite their efficacy, tumors may develop resistance to ICI, limiting the durability of the therapeutic benefit [11]. A major challenge in ICI is tumor heterogeneity, wherein genetic and molecular diversity within tumors leads to variable responses to treatment. As a result, specific drugs may only be effective against certain tumor subpopulations, complicating the treatment landscape [12].

To address these challenges, recent research suggests that the combination of various immunotherapeutic approaches or their integration with non-immunotherapeutic treatments may yield more effective solutions. This synergistic strategy holds the potential to enhance treatment efficacy, counteract resistance mechanisms, and extend the benefits of immunotherapy to a broader range of cancer types [8,13].

## 2. Mechanism of Action of Immunotherapy

Cancer immunotherapy enhances the body’s ability to detect and eliminate malignant cells by modulating key immune pathways. Unlike conventional therapies such as chemotherapy and radiation, which act directly on tumor cells, immunotherapy harnesses the adaptive immune system, primarily T cells, to recognize, target, and destroy cancer cells. This involves breaking immune tolerance, enhancing antigen presentation, and reversing immunosuppression within the tumor microenvironment (TME) [14].

At a molecular level, tumors often escape immune surveillance by exploiting regulatory checkpoints, molecular brakes that normally prevent autoimmunity. Two major checkpoints are Cytotoxic T-lymphocyte-associated protein 4 (CTLA-4) and programmed death–1 (PD-1)/Programmed death ligand 1 (PD-L1). CTLA-4 competes with the costimulatory molecule CD28 for binding to B7 on antigen-presenting cells (APCs), inhibiting early T cell activation in lymphoid organs. PD-1, on the other hand, is upregulated on T cells in the periphery and binds PD-L1 expressed on tumor cells or immune cells within the TME, leading to T cell exhaustion, impaired cytokine production, and reduced cytotoxicity [15,16].

Immune Checkpoint inhibitors, such as anti-PD-1, anti-PD-L1, and anti-CTLA-4 antibodies, block these inhibitory signals, thereby reactivating T cells and enhancing their ability to attack cancer cells. This strategy has led to remarkable improvements in cancer treatment, particularly in melanoma, non-small-cell lung cancer, and RCC, among others [17,18].

Beyond checkpoint inhibition, additional immunotherapeutic strategies are emerging to harness the immune system’s antitumor potential further. Among these, adoptive cell therapy (ACT) and cancer vaccines represent innovative approaches that aim to directly enhance immune cell targeting or train the immune system to recognize tumor-specific antigens. ACT is an advanced immunotherapy approach that involves extracting a patient’s T cells, genetically modifying them to enhance their tumor-targeting ability, and reinfusing them into the patient. One of the most effective forms of ACT is chimeric antigen receptor T-cell (CAR-T) therapy, where T cells are engineered to express synthetic receptors that recognize specific cancer antigens. CAR-T cells demonstrate potent tumor-killing ability and have been particularly successful in treating hematologic malignancies such as various subtypes of leukemia and lymphoma [19]. Cancer vaccines stimulate the immune system to recognize and destroy cancer cells by presenting tumor-associated antigens. These vaccines can be designed using dendritic cells, peptides, or viral vectors to train the immune system to mount an effective response against malignancies. Unlike preventive vaccines for infectious diseases, therapeutic cancer vaccines aim to enhance immune surveillance and eliminate existing cancer cells. Ongoing research is focused on developing personalized cancer vaccines tailored to an individual’s tumor-specific mutations [18].

Cytokines, such as interleukin-2 (IL-2), interferon-alpha (IFN-α), and granulocyte-macrophage colony-stimulating factor (GM-CSF), are essential mediators of the immune response against tumors. IL-2 therapy promotes the expansion and activation of cytotoxic T lymphocytes (CTLs) and natural killer (NK) cells, enhancing their tumor-killing capabilities. IFN-α and GM-CSF contribute to anti-tumor immunity by stimulating antigen presentation and increasing immune cell infiltration into tumors. Although historically used in metastatic RCC (mRCC), cytokine therapies such as IL-2 and IFN-α have largely fallen out of favor due to significant toxicity and limited efficacy. Their use has been supplanted by more effective and better-tolerated agents such as immune checkpoint inhibitors and VEGF-targeted therapies [1].

## 3. Preclinical Studies That Led to Immunotherapy

Preclinical research in immunotherapy for RCC has played a central role in shaping today’s treatment landscape. ccRCC, the most common histologic subtype, is recognized as an immunogenic tumor, largely due to its tendency to express PD-L1 and foster an immunosuppressive tumor microenvironment. This may help explain why some RCC tumors could evade immune detection despite being infiltrated by T cells [20].

In mouse models, blockade of the PD-1/PD-L1 and CTLA-4 pathways was shown to restore T-cell activity, enhance tumor killing, and delay disease progression, findings that directly supported early clinical trials of ICIs like nivolumab [21]. Another key insight from preclinical models was the immune-modulating effect of VEGF inhibitors. The clinical validation of immune checkpoint blockade began with the landmark CheckMate 025 trial, in which PD-1 inhibition with single-agent nivolumab significantly improved overall survival (OS) compared to everolimus in previously treated mRCC (median OS 25.0 vs. 19.6 months; HR 0.73), leading to the first FDA approval of nivolumab in RCC as a second-line therapy [22]. This success catalyzed further investigation into combination regimens for the first-line setting. Agents like sunitinib were shown not only to block angiogenesis but also to reduce myeloid-derived suppressor cells (MDSC) and improve T-cell infiltration, laying the groundwork for successful ICI-TKI combinations such as pembrolizumab + axitinib (KEYNOTE-426) and nivolumab + cabozantinib (CheckMate 9ER) [23,24].

In addition to checkpoint blockade, recent preclinical work has explored novel strategies such as neoantigen-based vaccines. A phase I trial demonstrated that personalized cancer vaccines targeting RCC-specific neoantigens could generate robust T-cell responses and potentially prevent tumor recurrence [25]. Meanwhile, studies have identified mechanisms of resistance to current therapies, including upregulation of alternative checkpoints like LAG-3 and TIM-3. To address this, dual ICI and cytokine-based approaches are under investigation to counteract immune escape and broaden response [26]. These preclinical insights have provided critical biological rationale for many of the combination regimens now tested in clinical trials and continue to guide the next generation of RCC immunotherapies.

## 4. Overview of Clinical Trials in RCC

The evolution of immunotherapy in RCC has shifted the treatment landscape from non-specific immune stimulation to precision-driven checkpoint blockade, offering durable outcomes for many patients.

### 4.1. Early Cytokine Therapies in RCC

The first major efforts in immunotherapy for RCC emerged from the observation that RCC was largely resistant to chemotherapy and radiotherapy, yet occasionally showed spontaneous regression, suggesting immune sensitivity [27]. Notably, some patients with metastatic disease who underwent nephrectomy experienced spontaneous resolution of pulmonary metastases. This phenomenon was attributed to the removal of the so-called “immunologic sink,” wherein the primary tumor was thought to suppress systemic anti-tumor immune responses, and its removal allowed for immune reactivation [28].

These insights led to the development of cytokine-based therapies, notably IL-2 and IFN-α [29]. High-dose IL-2, approved by the FDA in 1992, became the first immunotherapy for mRCC. IL-2 plays a critical role in activating lymphokine-activated killer (LAK) cells, which are immune cells derived from natural killer (NK) and T cells following several days of IL-2 stimulation. These LAK cells, which are part of the activated peripheral blood mononuclear cell (PBMC) population, exhibit potent cytotoxic activity against a broad spectrum of tumors while sparing normal tissues [30,31]. Early trials, including a pivotal study by Fyfe et al., demonstrated durable complete responses in approximately 7% of patients (17 out of 255) with mRCC treated with high-dose IL-2. However, its use was limited by high toxicity, including vascular leak syndrome, hypotension, and multi-organ failure, confining its administration to select patients at specialized centers [29]. Similarly, IFN-α was evaluated in multiple studies and showed modest benefits in progression-free survival (PFS) and overall survival (OS); however, its clinical impact was tempered by flu-like symptoms, fatigue, and psychiatric side effects [32].

### 4.2. Landmark Trials and the Turning Point in RCC Immunotherapy

The transition from cytokine therapy to ICIs marked a paradigm shift in RCC treatment. During this transitional period, targeted therapies particularly TKIs directed against the VEGF and mTOR pathways emerged as the dominant treatment approach, replacing cytokine therapy as the standard of care in advanced RCC [33]. Preclinical studies had identified PD-1/PD-L1 and CTLA-4 pathways as key mechanisms of immune escape in various solid tumors, including RCC [21]. This prompted the initiation of clinical trials targeting these checkpoints (Table 1). The CheckMate 025 trial was the first major trial to demonstrate the clinical potential of PD-1 inhibition in RCC. In this phase III study published in 2015, nivolumab significantly improved OS (25.0 vs. 19.6 months; HR 0.73; *p* = 0.002) compared to everolimus in previously treated patients with mRCC, establishing immune checkpoint blockade as a standard second-line therapy [22]. Importantly, the safety profile of nivolumab was more favorable than that of cytokines or mTOR inhibitors, with fewer grade 3–4 adverse events [22].

### 4.3. Advancements with ICI and Combinations

Following the success of CheckMate 025, efforts turned to earlier lines of therapy. The CheckMate 214 trial in 2018 evaluated nivolumab + ipilimumab, combining PD-1 and CTLA-4 blockade, in treatment-naïve mRCC patients. In those with intermediate- and poor-risk disease, the combination showed superior efficacy over sunitinib. Median OS was not reached in the combination group versus 26.0 months with sunitinib (HR 0.63), and median PFS was 11.6 months vs. 8.3 months, respectively (HR 0.73). The objective response rate (ORR) was also higher (42% vs. 27%). This trial established dual checkpoint inhibition as a viable first-line option in a biomarker-unselected population [34]. Updated 8-year follow-up data presented at ASCO 2024 confirmed the durability of this benefit, reporting a sustained OS hazard ratio of 0.75 in the same subgroup, along with a higher complete response rate of 11% compared to 2% with sunitinib [37].

Subsequently, ICI-based combinations with TKIs were investigated. KEYNOTE-426 (2019), a pivotal phase III trial, evaluated the efficacy of pembrolizumab, a PD-1 inhibitor, in combination with axitinib, a VEGFR tyrosine kinase inhibitor, compared to sunitinib in treatment-naïve patients with advanced RCC. A total of 861 patients were randomized to receive either pembrolizumab plus axitinib or sunitinib monotherapy. The combination demonstrated a 47% lower risk of death (hazard ratio for death, 0.53; 95% CI, 0.38–0.74; *p* < 0.0001), and a median PFS of 15.1 months compared to 11.1 months with sunitinib (HR 0.69; 95% CI, 0.57–0.84). The ORR was also higher in the pembrolizumab–axitinib arm (59.3%) than with sunitinib (35.7%), including a complete response rate of 10% versus 3.5%, respectively. These benefits were consistent across the International Metastatic RCC Database Consortium (IMDC) risk groups and PD-L1 expression subgroups [23]. Updated 5-year follow-up data presented at ASCO 2023 confirmed the durability of benefit, with a sustained OS advantage (HR 0.67; 95% CI, 0.52–0.84) and a 5-year OS rate of 41.9% versus 31.7%, favoring the pembrolizumab–axitinib combination [38].

This was followed by CheckMate 9ER, a pivotal phase III trial comparing the combination of nivolumab, a PD-1 inhibitor, and cabozantinib, a VEGFR tyrosine kinase inhibitor, against sunitinib monotherapy in treatment-naïve patients with advanced RCC in 2021. A total of 651 patients were randomized to receive either nivolumab–cabozantinib (*n* = 323) or sunitinib (*n* = 328). The combination regimen resulted in a significant improvement in PFS, with a median of 16.6 months versus 8.3 months for sunitinib (HR 0.51; 95% CI, 0.41–0.64; *p* < 0.001). Improvements were also seen in OS, with a 12-month OS rate of 85.7% for the combination arm versus 75.6% for sunitinib (HR 0.60; 98.89% CI, 0.40–0.89; *p* = 0.001). The ORR was notably higher with nivolumab–cabozantinib (55.7%) than with sunitinib (27.1%), including complete response rates of 8.0% and 4.6%, respectively. These clinical benefits were consistent across subgroups stratified by IMDC risk status and PD-L1 expression. Grade ≥ 3 treatment-related adverse events were reported in 60.6% of patients in the combination arm, compared to 50.9% in the sunitinib group [24]. Based on these findings, the combination of nivolumab and cabozantinib was approved as a first-line treatment option for advanced RCC.

The CLEAR trial (2021) further expanded first-line treatment options by evaluating lenvatinib, a multikinase inhibitor, in combination with pembrolizumab versus sunitinib in advanced RCC. In this phase III trial, 1069 treatment-naïve patients were randomized to receive either lenvatinib–pembrolizumab, lenvatinib–everolimus, or sunitinib. The lenvatinib–pembrolizumab arm demonstrated a significant improvement in PFS, with a median of 23.9 months compared to 9.2 months with sunitinib (HR 0.39; 95% CI, 0.32–0.49; *p* < 0.001). OS was also significantly improved (HR 0.66; 95% CI, 0.49–0.88; *p* = 0.005), and the ORR was 71.0%, including a complete response rate of 16.1%, compared to 36.1% and 4.2% with sunitinib, respectively. These results established lenvatinib–pembrolizumab as another highly effective first-line regimen for patients with advanced RCC, particularly for those with high disease burden or rapid progression [35].

## 5. Standard of Care

### 5.1. Localized and Locally Advanced RCC

Localized RCC, encompassing stages I and II, is primarily managed surgically, with the choice of procedure guided by tumor size, location, and patient-specific factors. For tumors ≤ 4 cm (T1a), partial nephrectomy is the preferred approach, aiming to preserve renal function while achieving oncologic control. Tumors measuring 4–7 cm (T1b) may still be amenable to partial nephrectomy, depending on their anatomical location and complexity; however, radical nephrectomy is often considered for larger or centrally located masses. Stage II tumors (>7 cm, T2) confined to the kidney are typically treated with radical nephrectomy [39].

Stage III RCC, classified as locally advanced, is defined by tumor extension into major veins (T3), perinephric tissues, or regional lymph node involvement (N1). It is primarily treated with radical nephrectomy. Lymph node dissection is selectively performed in patients with radiologically or intraoperatively suspicious lymphadenopathy but is not routinely indicated in the absence of nodal involvement [39].

Postoperative management decisions are influenced by the risk of recurrence (Table 2). The KEYNOTE-564 trial demonstrated that adjuvant pembrolizumab significantly improved disease-free survival and overall survival in patients with clear cell RCC at high risk of recurrence following nephrectomy, defined as those patients with stage II RCC with grade 4 or sarcomatoid features, stage III RCC, regional lymph node involvement (N+), or oligometastatic disease following metastatectomy (M1 NED) [40]. This led to FDA approval of pembrolizumab for adjuvant treatment in these patients. The S-TRAC trial demonstrated improved disease-free survival (DFS) with adjuvant sunitinib in high-risk clear cell RCC, but no benefit in OS, and its use remains limited due to toxicity [41]. Several other adjuvant strategies have failed to demonstrate benefit, including atezolizumab in the IMmotion010 trial, nivolumab plus ipilimumab in CheckMate 914, and axitinib in the ATLAS trial, which was terminated early due to futility [42,43,44].

### 5.2. Advanced/Metastatic Disease

Stage IV RCC, characterized by the presence of distant metastases, is managed primarily with systemic therapy. The current standard-of-care first-line treatment options fall into four major therapeutic classes: ICI combined with TKI, dual ICI therapy, TKI monotherapy (less commonly), and participation in clinical trials [39]. Selection among these regimens is largely guided by risk stratification based on the IMDC criteria. The IMDC model incorporates six clinical and laboratory factors: time from diagnosis to initiation of systemic therapy less than one year, Karnofsky performance status below 80%, anemia, hypercalcemia, neutrophilia, and thrombocytosis. Patients are categorized as favorable risk (0 risk factors), intermediate risk (1–2 factors), or poor risk (3 or more factors), and these classifications inform both prognosis and treatment selection [39,45].

For favorable-risk patients, ICI–TKI combinations are the preferred first-line approach. Three landmark trials have established these regimens: KEYNOTE-426, which evaluated pembrolizumab plus axitinib; CheckMate 9ER, which assessed nivolumab plus cabozantinib; and CLEAR, which assessed pembrolizumab plus Lenvatinib. All three combinations demonstrated superior OS, PFS, and ORR compared to sunitinib in previously untreated advanced RCC, including patients with favorable-risk disease [23,24,35]. These regimens are now widely adopted in this setting.

In patients with intermediate- or poor-risk disease, both ICI–TKI combinations and dual checkpoint blockade are approved options. The CheckMate 214 trial evaluated the combination of nivolumab and ipilimumab, demonstrating a significant OS benefit and a higher rate of durable complete responses compared to sunitinib in intermediate- and poor-risk groups [34]. This dual ICI regimen remains a standard first-line option in patients, especially those with sarcomatoid histology or those for whom treatment-free intervals are a clinical goal. ICI–TKI combinations also remain appropriate alternatives in this population [23,24].

In the second-line setting, treatment selection is guided by the patient’s prior exposure and tolerability to first-line therapy. With the increasing adoption of ICI-based regimens as the standard of care in the frontline setting, the use of nivolumab monotherapy in the second-line therapy has become relatively uncommon. Most patients who do not receive ICI up front typically have a contraindication, such as a history of solid organ transplantation or uncontrolled autoimmune or rheumatologic disease, that precludes immunotherapy altogether [39,46]. In such cases, sequential use of VEGF-targeted TKIs remains the primary approach [39].

For patients previously treated with ICI-TKI combinations, second-line options include cabozantinib, lenvatinib plus everolimus, or other VEGF inhibitors [36,46,47]. For the small subset of patients who received frontline VEGF monotherapy, ICIs such as nivolumab may still be considered, as supported by results from the CheckMate 025 trial, which demonstrated improved OS compared to everolimus [22].

In the third-line setting and beyond, therapeutic decision-making is individualized and largely influenced by prior lines of treatment, disease burden, and patient comorbidities. Agents such as tivozanib, axitinib, and lenvatinib plus everolimus remain viable options if not previously utilized [39,47,48,49,50]. Belzutifan, a hypoxia-inducible factor 2-alpha (HIF-2α) inhibitor that targets the VHL–HIF pathway, has demonstrated promising efficacy in patients with VHL-associated RCC [51]. The phase II LITESPARK-005 trial examined belzutifan in the ccRCC setting and found that belzutifan significantly improved PFS compared to everolimus in previously treated patients with advanced ccRCC, supporting its potential role as a non-immunologic, molecularly targeted therapy in later lines [52]. As the therapeutic landscape continues to evolve, clinical trial participation remains strongly encouraged to expand treatment options and refine sequencing strategies.

## 6. Role of Biomarkers

PD-L1 is a Programmed Cell Death Ligand that acts as a roadblock to T cell activation and helps the tumor cells evade the immune system. Although it is an important predictive factor for response to immunotherapy in NSCLC [53], in RCC, PD-L1 is not used as a predictive factor. IO-TKI combinations are used irrespective of the PD-L1 expression in clinical trials such as Checkmate 214 and Keynote 426 [23,34].

Tumor Mutational burden (TMB) is the number of somatic mutations per unit of tumor DNA. Its utility in RCC as a biomarker for response to Immunotherapy is limited. Studies such as Checkmate 025 and Checkmate 9ER failed to identify any consistent correlation between TMB and response to IO [22,24,54].

TIM-3 expression is being increasingly studied in RCC and is not established, but it is an emerging biomarker in RCC. It is often co-expressed in CD8 T cells with PD-L1, leading to T cell exhaustion and an Immunosuppressive tumor milieu [55]. High TIM-3 expression in RCC shows poor response to Immunotherapy and poor prognosis [56]. Co-blockade of PD-L1 and TIM-3 is currently being studied in RCC to overcome resistance and improve efficacy of Immunotherapy [57].

LAG-3 is another biomarker. It can be co-expressed on the CD8 T cells, causing T cell dysfunction and an immunosuppressive tumor microenvironment [58]. It is not used as a predictive marker alone in RCC, but it is associated with poor prognosis and decreased overall survival in RCC [59]. It is currently being explored as a therapeutic target in RCC in combination with PD-1 blockers [60].

The Gut Microbiome is also being studied as a predictive marker for response to Immunotherapy in RCC. Studies have shown that responders have high levels of immune stimulatory bacteria like Akkermansia muciniphila, Bifidobacterium, and Faecalibacterium, while non-responders showed depletion of these species [61,62].

Other than the above-mentioned predictive markers, IMDC risk stratification in RCC was developed using several adverse prognostic markers, which were initially designed to predict response to VEGF-based therapy. It has become an essential measure in treatment selection and response prediction in the era of immunotherapy [45]. CheckMate 214 showed that RCC patients with Intermediate to Poor risk groups showed better response to dual ICI vs. Sunitinib [34]. In the favorable risk group, Sunitinib showed better response than the dual ICI combination. ICI+VEGF-TKI combinations are active across all groups, including the favorable risk group [23].

## 7. ICI Resistance and Escape Mechanisms

Like other treatments, resistance to ICI remains a major clinical challenge. Once resistance develops, current standard second- and third-line therapies offer limited benefits in terms of PFS and OS [47,52]. To understand resistance, it is important to first review the basic steps of the anti-tumor immune response triggered by ICI. The process begins with dendritic cells presenting tumor-associated antigens to T lymphocytes, enabling recognition of cancer cells. These activated T cells then migrate into the tumor microenvironment, where they are further activated and secrete cytokines. These cytokines recruit additional immune cells, such as macrophages and dendritic cells, to sustain the immune attack. Finally, the immune system forms memory T cells that help maintain long-term surveillance against the tumor [14,63].

Resistance to ICIs in RCC can be broadly categorized as primary, adaptive, or acquired (Table 3). Primary resistance refers to a lack of clinical benefit from immunotherapy despite adequate drug exposure and no apparent pharmacologic failure. It is generally defined as disease progression or lack of response (per RECIST 1.1 criteria) at the first radiographic evaluation, usually within 8–12 weeks of initiating ICI therapy [14,63,64]. Mechanisms include an inherently non-immunogenic tumor microenvironment, low tumor mutational burden, or absence of T-cell infiltration (also known as “cold tumors”) [14,65].

Adaptive resistance occurs when tumors initially trigger immune activation but subsequently upregulate immune checkpoint molecules (e.g., PD-L1) or immunosuppressive pathways in response to immune pressure. This dynamic immune evasion allows tumors to escape destruction after initial immune recognition [82].

Acquired resistance describes cases where a patient initially responds to immunotherapy but later experiences disease progression after a period of tumor control. Mechanisms may include loss of neoantigen expression, immune editing, or T-cell exhaustion [83].

These resistance patterns may stem from tumor-intrinsic factors (e.g., genomic mutations, antigen presentation defects), host-related factors (e.g., T-cell repertoire, HLA type), or extrinsic influences within the tumor microenvironment (e.g., immunosuppressive cytokines, Tregs) [83].

### 7.1. Tumor-Related Factors for Resistance to ICI

Tumor-related mechanisms of ICI resistance are multifaceted and begin at the level of antigen recognition. Effective immune surveillance requires that dendritic cells present tumor-associated antigens to T cells, a process that can be disrupted when tumors are poorly differentiated or undergo dedifferentiation. Such changes may lead to loss or mutation of critical cell surface molecules, including major histocompatibility complex class I (MHC-I) and β2-microglobulin, impairing antigen presentation. Mutations or dysregulation in signaling pathways, such as the interferon (IFN), MAPK, and JAK/STAT pathways, can further diminish tumor visibility to the immune system [60]. Stress within the endoplasmic reticulum also interferes with antigen processing, while mutations in mitochondrial cytochrome proteins may disrupt apoptotic signaling. Aberrant expression of PD-L1, whether upregulated as an immune evasion tactic or downregulated in certain resistant phenotypes, influences tumor susceptibility to checkpoint blockade. Additionally, epigenetic reprogramming can alter the expression of immune-related proteins even in the absence of genetic mutations. The tumor microenvironment itself can become immunosuppressive through the secretion of inhibitory metabolites and cytokines, creating a barrier to effective T cell infiltration and activation. Resistance to apoptosis further enables tumor cells to evade immune destruction. Epithelial-to-mesenchymal transition (EMT), a key process in tumor progression, has also been implicated in immunotherapy resistance through downregulation of immune checkpoints like PD-L1, ultimately contributing to immune evasion and therapeutic failure [84,85,86].

### 7.2. External Factors in the Tumor Microenvironment That Lead to Resistance to ICI

When tumor-infiltrating lymphocytes (TILs) penetrate the tumor microenvironment, some may differentiate into Tregs, which suppress the antitumor immune response and promote tumor progression [87]. This immunosuppressive shift is further influenced by alterations in cytokine signaling. Elevated levels of immunosuppressive cytokines such as TGF-β and IL-2 facilitate Treg development and inhibit cytotoxic T cell function [88,89]. Chemokines, which orchestrate the recruitment of immune cells, also play a dual role in tumor biology. While chemokines such as CXCL9 and CXCL10 are associated with antitumor immunity through the recruitment of effector T cells, others like CCL2 and IL-8 exhibit pro-tumorigenic effects [90,91,92]. These pro-tumor chemokines promote the infiltration of MDSCs, which in turn suppress cytotoxic T lymphocytes and secrete tumor-promoting growth factors such as TGF-β, fibroblast growth factor (FGF), platelet-derived growth factor (PDGF), and epidermal growth factor (EGF), ultimately enhancing tumor growth and immune evasion [93].

VEGF secreted by tumor cells can lead to abnormal blood vessel formation, leading to stress in the TME because of the increased oxygen requirement by the tumor, leading to hypoxia and acidosis [94]. Hypoxia contributes to an immunosuppressive tumor microenvironment by inducing HIF-1α in both tumor cells and MDSCs, a heterogeneous population of immature myeloid cells that suppress T cell responses, resulting in increased PD-L1 expression, T cell depletion, and resistance to ICIs [95]. Additionally, HIF-1α upregulates VEGF expression through a positive feedback loop, further promoting immune evasion and tumor progression [96]. VEGF can also bind to dendritic cells (DCs) and stop its maturation process, thereby impairing antigen presentation and subsequent T cell activation [97]. Abnormal blood vessel formation causes difficulty in infiltration of T cells because of the alterations in adhesion molecules like vascular cell adhesion molecule (VCAM), *p* selectin, E selectin, Platelet endothelial cell adhesion molecule-1 (PECAM-1), CD31, and CD99 [98]. Furthermore, these changes in vasculature can inhibit penetrance to the tumor itself [99].

### 7.3. Alterations in Cell Metabolites and Effects on Immune Response

High lactate levels caused by hypoxia result in the expression of CD44 and hyaluronic-associated metastasis [100]. Tumor cells increase CD38 expression to produce adenosine, leading to tumor metastasis and causing increased expression of PD-1 on the T cells and impaired signaling [101]. Additionally, tumor cells and MDSCs utilize tryptophan and form an immunosuppressive metabolite kynurenine [102]. Tumor cell esterification of cholesterol can lead to maturation of inhibitory T cells, and lead to deactivation DCs [100].

## 8. Future of IO in RCC

Several ICI–TKI combinations are approved for patients across all IMDC risk groups, along with dual checkpoint blockade with nivolumab and ipilimumab for intermediate- and poor-risk groups. Ongoing research is focused on refining existing strategies and introducing novel immunotherapeutic approaches. Among emerging agents, LAG-3 + PD-1 inhibitors are under investigation for PD-L1 low tumors, while additional immune checkpoints such as TIGIT, TIM-3, KIR3DL3, and PSGL-1 are also being explored as potential targets [103,104,105]. Cytokine-based therapies, including CD8-IL-2 fusion proteins and decoy-resistant IL-18, aim to rejuvenate exhausted T-cell pools and have shown promising in vitro responses [106,107]. Personalized mRNA-based cancer vaccines using tumor-specific neoantigens have demonstrated encouraging recurrence-free survival beyond 40 months in resected stage III and IV RCC in early-phase trials [25]. Cell-based therapies such as CAR-T targeting CD70 (e.g., CTX130 and ALLO-316) have also shown potential in early studies of advanced or refractory RCC [108,109]. Utilizing the gut microbiome, the addition of Clostridium Butyricum improves the response from immunotherapy, showing significantly longer PFS in early phase clinical trials [110].

Importantly, several recent and ongoing clinical trials may shift practice. The Phase III LITESPARK-005 trial led to the FDA approval of belzutifan in 2023 after demonstrating a higher ORR (22% vs. 4%) and a significantly reduced risk of progression or death compared to everolimus (HR 0.75; *p* = 0.0008) in pretreated ccRCC [52]. Rechallenge strategies have thus far not shown benefit, with both the CONTACT-03 study [111] (IO rechallenge with atezolizumab) and the TiNivo-2 study [112] (IO rechallenge with nivolumab) showing lack of benefit. Adaptive trial designs, such as the ongoing PEDIGREE trial, tailor post-induction therapy based on early responses to combination treatment (nivolumab and ipilimumab), allowing for the potential early addition of cabozantinib, aiming to improve outcomes through a more personalized approach [113]. Additionally, the PROBE trial (NCT04510597) is evaluating the role of cytoreductive nephrectomy in patients with metastatic disease receiving IO-based systemic therapy, the results of which are eagerly anticipated [114].

## 9. Conclusions

Immunotherapy has revolutionized the treatment landscape of RCC, offering durable responses and improved survival outcomes, particularly in advanced and metastatic settings. While checkpoint inhibitors and combination regimens have become integral to standard care, challenges such as resistance, immune evasion, and tumor heterogeneity continue to limit long-term efficacy. Ongoing research into novel immune targets, personalized vaccines, CAR-T therapies, and modulation of the tumor microenvironment holds promise for enhancing and extending the benefits of immunotherapy. Continued innovation and a deeper understanding of RCC immunobiology will be key to developing more effective, personalized, and durable treatment strategies in the future.

## Figures and Tables

**Table 1 cancers-17-03139-t001:** Notable Clinical Trials Related to RCC Immunotherapy.

Trial Name	Trial ID	Phase	Objective	Number of Patients	Median Follow up	Results
CheckMate 025 [22]	NCT01668784	III	Nivolumab vs. Everolimus in previously treated metastatic RCC	406 vs. 397	14 months	OS: 25.0 vs. 19.6 months (HR 0.73; *p* = 0.002); better safety profile
KEYNOTE-426 [23]	NCT02853318	III	Pembrolizumab + Axitinib vs. Sunitinib in first-line advanced RCC	432 vs. 429	12.8 months	Median PFS: 15.1 vs. 11.1 months (HR 0.69; *p* < 0.001); OS: HR 0.53 (*p* < 0.0001); ORR: 59.3% vs. 35.7%
CheckMate 9ER [24]	NCT03141177	III	Nivolumab + Cabozantinib vs. Sunitinib in first-line metastatic RCC	323 vs. 328	18.1 months	Median PFS: 16.6 vs. 8.3 months (HR 0.51; *p* < 0.001); OS at 12 months: 85.7% vs. 75.6% (HR 0.60; *p* = 0.001); ORR: 55.7% vs. 27.1%
CheckMate 214 [34]	NCT02231749	III	Nivolumab + Ipilimumab vs. Sunitinib in treatment-naive metastatic RCC	547 vs. 535	25.2 months	OS: HR 0.63 (*p* < 0.001); ORR: 42% vs. 27%; more benefit in intermediate/poor-risk patients
CLEAR Trial [35]	NCT02811861	III	Lenvatinib + Pembrolizumab or Everolimus vs. Sunitinib in first-line advanced RCC	355 vs. 357 vs. 357	26.6 months	Median PFS: 23.9 vs. 9.2 mo (HR 0.39; *p* < 0.001); OS: HR 0.66 (*p* < 0.001); ORR: 71% vs. 36%
JAVELIN Renal 101 [36]	NCT02684006	III	Avelumab + Axitinib vs. Sunitinib in first-line advanced RCC	442 vs. 444	9.9 months in Avelumab + Axitinb; 8.4 months in Sunitib	Median PFS: 13.8 vs. 8.4 months (HR 0.69; *p* < 0.001); OS not significantly improved; ORR higher in combination arm
Phase I Trial of Personalized Cancer Vaccines in RCC	NCT03472238	I	Personalized cancer vaccines targeting RCC-specific neoantigens			Robust T-cell responses observed; early-phase safety and immunogenicity data encouraging
Combination of Dual Checkpoint Inhibition (PD-1/PD-L1 + LAG-3/TIM-3)	NCT03871297	I/II	PD-1/PD-L1 + LAG-3/TIM-3 inhibitors in resistant advanced RCC			Ongoing trial; aims to overcome resistance with novel checkpoint blockade combinations

**Table 2 cancers-17-03139-t002:** Adjuvant Treatment Studies.

Trial	Therapy	Population	Outcome	Result
S-TRAC [41]	Sunitinib	High-risk (≥pT3 and/or N+)	Improved DFS; no OS benefit	Positive (DFS)
KEYNOTE-564 [40]	Pembrolizumab	Intermediate-high/high-risk	Improved DFS and OS	Positive
CheckMate 914 [34]	Nivolumab + Ipilimumab	High-risk	No DFS benefit	Negative
IMmotion010 [42]	Atezolizumab	High-risk	No DFS benefit	Negative
ATLAS [44]	Axitinib	High-risk	Trial stopped early; no benefit	Negative

**Table 3 cancers-17-03139-t003:** Mechanisms of Resistance to Immunotherapy in RCC.

Type of Resistance to IO	Factors/Mechanism	Description	Therapeutic Intervention
Primary Resistance to ICI	Poor Antigenicity/Tumor differentiation	Poor Tumor differentiation and Lack of Neoantigens leads to defective antigen presentation and poor T cell recognition	Vaccines DNMT/HDAC inhibitors to enhance Neoantigen expression [66,67]
	Loss of MHC-I/Beta Microglobulin	Tumor cells may lose MHC-I expression, impairing antigen presentation to T cells.	NK cell-based therapies that target MHC-I-deficient tumors [68]
	Signaling alterations in pathways (INF, MAPK, JAK-STAT)	Changes in signaling pathways can impact immune recognition by tumor cells	Target downstream pathways or restore IFN signaling [66,69]
	WNT/β-catenin pathway activation	Activation of β Catenin leads to suppression of T cell infiltration	WNT inhibitors are being studied in RCC to restore the T cell infiltration [70]
	PTEN loss → PI3K pathway activation	PI3K activation leads to suppression of T cell infiltration	Combining ICI with PI3K inhibitors [71]
Secondary Resistance to ICI	Acquired B2M or MHC-I mutations	Leads to defects in Antigen presentation to T cells	NK cell based or engineered TCR therapies [68]
	Acquired JAK1/2 mutations post treatment	Signaling alterations leads to poor recognition by T cells	STING agonists and Adaptive T cell therapy [66,69]
	Upregulation of alternate checkpoints (TIM-3, LAG-3)	Upregulation of alternative check points leads to immune evasion and resistance to ICIs	Anti TIM-3 Antibodies alone or combined with PD-1 inhibitors is being studied [72,73]
	WNT/PI3K activation post treatment	Activation of either of these pathways can lead to decreased T cell infiltration	Dual blockade of both the pathway downstream inhibition combined with PD1 [74]
Tertiary Resistance	T regulatory cells infiltration	Leads to Immuno suppressive tumor environment	Treg depletion (low-dose cyclophosphamide) [75]
	MDSCs infiltration	MDSCs directly suppress the CD 8 + T cells and CD 4+ T cells decreasing the anti Tumor immunity	MDSC inhibitors (CSF1R blockers) [76]
	VEGF-mediated hypoxia & acidosis	Abnormal blood vessel proliferation leading to hypoxia and upregulation of PD-L1 leading to immunosuppressive environment	Combine ICI with VEGF inhibitors [77]
	Metabolic suppression through increase in Indolamine 2,3 diaoxygenase, adenosine, CD38	Causes impaired T cell function leading to increase in Immune evasion	IDO inhibitors (e.g., epacadostat), CD38 blockers, adenosine A2A receptor inhibitors [78,79,80]
	Lactate accumulation, CD 44 expression	Associated with immune evasion in tumor with high glycolysis tumors like RCC	Target glycolysis (LDH inhibitors), CD44 blocking agents [81]

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
