# Peer review of "A Review of Immunotherapy in Renal Cell Carcinoma: Current Landscape and Future Directions"

_cancers, 2025, doi:10.3390/cancers17193139_

Round 1

Reviewer 1 Report

Comments and Suggestions for Authors

This manuscript is a valuable contribution to the field of uro-oncology and offers a state-of-the-art synthesis of immunotherapy in RCC. With minor adjustments, it will serve as a highly informative resource for clinicians.

Recommendations for Improvement:

  1. Table 1 Improvements:
    • Include patient numbers for each trial arm.
    • Clearly specify follow-up duration for each analysis presented.
  2. Biomarkers and Predictive Factors:
    • Insert a separate and detailed discussion section regarding biomarkers and predictive factors. This should include PD-L1 expression, tumor mutational burden, emerging biomarkers such as LAG-3, TIM-3 expression, gut microbiome signatures, and the role of IMDC and other risk stratifications.
  3. Table 3 Enhancements:
    • Add an additional column detailing potential therapeutic interventions for each listed resistance mechanism.
    • Include frequency or prevalence data for each resistance mechanism, where such information is available.
    • Consider organizing resistance mechanisms clearly by primary, secondary, and tertiary resistance patterns.
    • Add relevant references to substantiate each resistance mechanism discussed.
  4. Treatment Algorithm:
    • Consider adding a clear and practical treatment algorithm figure summarizing therapeutic approaches based on different clinical scenarios and resistance patterns.

Author Response

Reviewer 1

Comment 1: Table 1 Improvements:

    • Include patient numbers for each trial arm.
    • Clearly specify follow-up duration for each analysis presented.

Response: The following changes have been made in Table 1 to include the number of patients and the follow-up duration.

Comment 2: Biomarkers and Predictive Factors:

    • Insert a separate and detailed discussion section regarding biomarkers and predictive factors. This should include PD-L1 expression, tumor mutational burden, emerging biomarkers such as LAG-3, TIM-3 expression, gut microbiome signatures, and the role of IMDC and other risk stratifications.

Response: We have added a whole section addressing this on page 8, line 352.

Comment 3: Table 3 Enhancements:

    • Add an additional column detailing potential therapeutic interventions for each listed resistance mechanism.
    • Include frequency or prevalence data for each resistance mechanism, where such information is available.
    • Consider organizing resistance mechanisms clearly by primary, secondary, and tertiary resistance patterns.
    • Add relevant references to substantiate each resistance mechanism discussed.

Response: Table 3 has been modified to include all the changes.

Comment 4: Treatment Algorithm:

    • Consider adding a clear and practical treatment algorithm figure summarizing therapeutic approaches based on different clinical scenarios and resistance patterns.

Response: We were unable to create a figure with a treatment algorithm based on clinical scenarios.

Reviewer 2 Report

Comments and Suggestions for Authors

In the article titled, ‘A Review of Immunotherapy in Renal Cell Carcinoma: Current Landscape and Future Directions’ authored by Peshin et.al., have systematically summarized the recent immunotherapeutic developments in RCC. The authors have also explored the advancement evolution of cytokine-based treatments, mRNA vaccines, CAR T approaches, and involvement of gut microbiome to the advent of immune checkpoint inhibitors (ICIs) and their integration with receptor tyrosine kinase inhibitors (TKIs).  They have included key studies such as CheckMate 214, KEYNOTE-426, and CheckMate 9ER that have played important roles in the development of immuno-oncology based therapies.

The topic is interesting and have potential however, there are several weakness in the writing

All the section, figures need significant improvements, some of them are listed below.

Writing needs coherence, everything needs to be defined when first introduced, connections are very weak.

New molecules are introduced without any logic

In the introduction section, authors have provided too much background information on RCC, its subtypes and its classification. This information is available in several dedicated article (https://jamanetwork.com/journals/jama/fullarticle/2822917 ; https://www.nature.com/articles/nrdp20179 ; ), here authors should only provide brief background of RCC and include introduce more on immunotherapeutic approaches in RCC and how it could be better than tyrsoine kinase inhibitors, radiation and chemotherapeutic approaches.

Section 2.1-2.1.4  The sections are superficially written and I am not sure what is the use of all of this information, it does fit with the heading ‘Mechanism of Action of Immunotherapy: A Detailed Overview’. Why authors want to discuss ‘Allergen-Specific Immunotherapy (AIT)’ and how it is related to ‘Immunotherapy in Renal Cell Carcinoma’ ?

Authors have discussed Cancer Vaccines (section 2.2.3) under Cancer immunotherapy section (section 2.2),  why ? Also, the whole section is written superficially and lacks details and molecular pathways involved in immunotherapy

Define OS in line 245 as  authors did for PFS

In section 4.1 authors should include the year when the study was initiated with the following statement (line 254) ‘The CheckMate 025 trial was the first major trial to…’ (include year’s wherever possible, this will provide a timeline to the readers)

Why did authors decided to include a figure with CAFs? Though they have not discussed/introduced CAFS anywhere in the text ?

Current figures are poor and authors needs to upgrade to include more figures, such as a figure showing immunotherapeutic approaches/targets in RCC.  Authors could have included more figures, such as  for section 6 ‘ICI Resistance and Escape Mechanisms’

Author Response

Reviewer 2:

Comment 1: In the introduction section, authors have provided too much background information on RCC, its subtypes and its classification. This information is available in several dedicated article (https://jamanetwork.com/journals/jama/fullarticle/2822917 ; https://www.nature.com/articles/nrdp20179 ; ), here authors should only provide brief background of RCC and include introduce more on immunotherapeutic approaches in RCC and how it could be better than tyrsoine kinase inhibitors, radiation and chemotherapeutic approaches.

Response: We shortened the introduction section and provided only a brief background. We included more about immunotherapy approaches and its comparison with Tyrosine Kinase inhibitors, radiation and chemotherapy approaches in paragraph 3, 4 and 5 of introduction, on page 3.

Comment 2: Section 2.1-2.1.4  The sections are superficially written and I am not sure what is the use of all of this information, it does fit with the heading ‘Mechanism of Action of Immunotherapy: A Detailed Overview’. Why authors want to discuss ‘Allergen-Specific Immunotherapy (AIT)’ and how it is related to ‘Immunotherapy in Renal Cell Carcinoma’ ?

Response: The entire section 2 has been rewritten, we have omitted the part of allergen specific immunotherapy.

Comment 3: Authors have discussed Cancer Vaccines (section 2.2.3) under Cancer immunotherapy section (section 2.2),  why ? Also, the whole section is written superficially and lacks details and molecular pathways involved in immunotherapy

Response: We agree that cancer vaccines have not yet demonstrated significant clinical efficacy in RCC and remain largely investigational. However, we have retained a concise discussion of this strategy to provide a complete overview of immunotherapeutic modalities under exploration, particularly as neoantigen-based personalized vaccines are beginning to show immunogenic potential in early-phase RCC trials. This inclusion offers readers insight into emerging strategies that may complement or synergize with existing immune checkpoint blockade in the future.

Comment 4: Define OS in line 245 as  authors did for PFS

Response: OS has been defined on page 5, line 153.

Comment 5: In section 4.1 authors should include the year when the study was initiated with the following statement (line 254) ‘The CheckMate 025 trial was the first major trial to…’ (include year’s wherever possible, this will provide a timeline to the readers)

Response: We have added years in section 4 to provide a timeline to the readers.

Comment 6: Why did authors decided to include a figure with CAFs? Though they have not discussed/introduced CAFS anywhere in the text ?

Response: We have removed that figure.

Comment 7: Current figures are poor and authors needs to upgrade to include more figures, such as a figure showing immunotherapeutic approaches/targets in RCC.  Authors could have included more figures, such as  for section 6 ‘ICI Resistance and Escape Mechanisms’

Response: Those figures have been removed. We are unable to create accurate figures to match ICI resistance and escape mechanism.

Round 2

Reviewer 2 Report

Comments and Suggestions for Authors

The authors have satisfactorily revised the manuscript.